# Evolution of Microstructure and Elements Distribution of Powder Metallurgy Borated Stainless Steel during Hot Isostatic Pressing

**Yanbin Pei** [1,2,3] **, Xuanhui Qu** [2,]* **, Qilu Ge** [1,]* **and Tiejun Wang** [4]

1 Central Iron & Steel Research Institute, Graduate School, Beijing 100081, China; peiyanbin@atmcn.com
2 Institute for Advanced Materials and Technology, University of Science and Technology Beijing, Beijing 100083, China
3 Antai-Heyuan Nuclear Energy Technology & Materials Co., Beijing 100094, China
4 Advanced Technology & Materials Co., Ltd., Beijing 100081, China; wangtj@atmcn.com
* Correspondence: quxh@ustb.edu.cn (X.Q.); geqilu@crisi.com.cn (Q.G.);
  Tel.: +86-10-6233-2700 (X.Q.); +86-10-6218-2806 (Q.G.)

**Abstract:** Prepared by powder metallurgy process incorporating atomization and hot isostatic pressing (HIP) sintering at six different temperatures from 600 to 1160 °C, borated stainless steel (BSS) containing boron content of 1.86 wt% was studied. The phase of BSS, relative density of different temperature, microstructure, elemental distribution, and mechanical properties were tested and analyzed. The phases of the alloy were calculated by the Thermo-Calc (2021a, Thermo-Calc Software, Solna, Sweden) and studied by quantitative X-ray diffraction phase analysis. The distributions of boron, chromium, and iron in grains of the alloy were analyzed by scanning electron microscopy and transmission electron microscope. The grain size distributions and average grain sizes were calculated for the boron-containing phases at 900, 1000, 1100, and 1160 °C, as well as the average grain size of the austenite phase at 700 and 1160 °C. After undergoing HIP sintering at 900, 1000, 1100, and 1160 °C, respectively, the tensile strength and ductility of the alloy were tested, and the fracture surfaces were analyzed. It was found that the alloy consisted of two phases (austenite and boron-containing phase) when HIP sintering temperature was higher than 900 °C, and the relative density of the prepared alloys was higher than 99% when HIP temperature was higher than 1000 °C. According to the boron-containing phase grain size distribution and microstructure analysis, the boron-containing phase precipitated both inside the austenite matrix and at the grain boundaries and its growth mechanism was divided into four steps. The tensile strength and elongation of alloy were up to 776 MPa and 19% respectively when the HIP sintering was at 1000 °C.

**Keywords:** borated stainless steel; different hot isostatic pressing sintering temperature; microstructure; boron distribution; powder metallurgy; boron-containing phase growth mechanism; Thermo-Calc Software

## 1. Introduction

Materials containing element boron are capable of absorbing thermal neutrons due to the high neutron cross-section of isotope $^{10}$B [1]. Steel containing more than 0.1% boron is excellent neutron absorber material. The 18Cr-8Ni austenitic stainless steel containing boron has higher mechanical properties and corrosion resistance [2], so American Society of Testing Materials (ASTM) issued standard A 887 (2004, ASTM International, West Conshohocken, PA, USA), which covers chromium-nickel stainless steel plate, sheet, and strip for nuclear application [3]. Both powder metallurgy (PM) [4–6] and casting processes [7–9] can produce borated stainless steel (BSS) alloys, but the former has more uniform distribution of boron and higher mechanical properties than the latter.

It is normally considered that the element boron is not easily soluble in Fe–Cr steels, the solubility value was only 0.004 to 0.008 mass% was when tested at 900 °C and dropped

to less than 0.0005 mass% at 700 °C in austenite and ferrite steel [10,11]. Gas atomization of molten alloy can produce the supersaturated solid solution due to high quenching rates [12–15]. Produced by atomization the BSS powder is supersaturated solid solutions at room temperature. At higher temperatures, boron will precipitate and decompose from solution. It is generally accepted that the decomposition and precipitation of supersaturated solid solutions can be divided into three separate steps according to kinetics [16]: the new phase nuclei forms in the first step; supersaturation decreases while precipitation bulk density remains constant in the second step and the precipitations coalesce and grow in size while quantity decrease in the third step.

The PM (powder metallurgy) process is better than the casting process to prepare BSS with more uniform distribution of boron elements, finer grains, and higher mechanical properties. Hot isostatic pressing (HIP) sintering is the process to densify powders or cast and sintered parts in the furnace under both high pressure (100–200 MPa) and high temperatures for example special steels, superalloys, ceramics etc. [17]. The temperature of the powder sintering by HIP is generally greater than $0.7T_m$ ($T_m$: melting point) [18]. The pressure of HIP (typically approximately 100 MPa) is much greater than the driving force of closure among powder [18], therefore dense materials by HIP can be obtained at a lower temperature than pressureless sintering due to the combined effect of temperature and pressure [19]. The densification of the powder occurs mainly during the step of raising and holding temperature and pressure. The powder in most region flows centrally in the powder densification, but there is a complex flow around the edge of capsule, so that the final HIP sintering may be somewhat uneven in density [20]. The material, which is made from HIP densification of gas atomized have smaller and more homogeneous grains and have higher tensile, strengths, ductility, and toughness than conventionally processed material [21,22]. Oxygen element cannot be reduced during HIP sintering which is harmful element that can lead to small pores in the microstructure and there is a linear relationship between the oxygen content and the area fraction of the pores [23]. Therefore, it is necessary to minimize the oxygen content of the powder to improve the properties of HIP sintering materials. A number of process parameters and material properties of powdered stainless steel were investigated after HIP [24–28].

HIP sintering of BSS process is boron precipitation from supersaturated solid solution, segregation, boron diffusion and boron-containing phase generation process. The grain boundary segregation behavior of boron in austenitic stainless was mainly of the non-equilibrium type [29]. The nonequilibrium is greatly influenced by the cooling rate and the nonequilibrium segregation of boron at grain boundaries in austenite is a kinetic phenomenon [30]. At higher temperatures, longer diffusion distances and more excess vacancies can enhance the nonequilibrium segregation of boron [31]. Boron diffusing rate in steel is similar to carbon, and promptly fast cooling rates (about 500 °C/s) and lower temperatures do not limit the diffusion of boron [32]. Boron diffusion mechanism is complex for non-equilibrium segregation. Using first-principles calculations, boron diffusion in faced centered cubic (fcc) Fe was the Boron–monovacancy mechanism and the binding energy of boron with monovacancy was 0.20 eV [33].

Previous work [5] by our team investigated coupons that were prepared by HIP at 1100 °C, and then conducted heat treatment ranging 900 to 1200 °C. We found that the alloy phases consisted with austenite and $Fe_{1.1}Cr_{0.9}B_{0.9}$ phase after heat treatment carrying out. Due to the different diffusion coefficients of Cr, Fe and B at distinct temperature, the distribution of the elements in the BSS was not uniform. When the heat treatment temperature of the alloy remains at 1000–1150 °C, the tensile strength and elongation were approximately 800 MPa and 20%. It can be concluded from this work that the suitable heat treatment temperatures (1000–1150 °C) can improve the mechanical properties of BSS by HIP, due to the different elemental distribution from the unheated treatment alloy without heat treatment.

To further improve the mechanical properties of BSS, we believe that it is necessary to improve the properties of HIP billet ingots prior to heat treatment in addition to select the

optimum heat treatment process. From the microscopic point of view, there are two ways to achieve the improvement of the property of HIP billets. One is to refine the grain of boride, to play the role of dispersion strengthened, so lowering the HIP sintering temperature is one way to be expected; the other is to improve the bonding of grain boundaries, increasing the temperature may be achieved. So, the HIP temperature chosen was 900–1160 °C; if the temperature was any higher, low melting point eutectics may appear and the borides grow to affect property. The research goal was to analyze the microstructure mechanisms of BSS in HIP processes. The work objective was to determine the optimum HIP sintering temperature to obtain better mechanical properties.

In HIP densification of BSS, critical factors to improve the material properties, include phase and microstructure evolution, the distribution of boron and other elements, density changes with temperature, but there are few published reports. In this study, the generation mechanism of the boron-containing phase in BSS can be inferred from the test and analysis of phase, microstructure, grain size and elemental distribution at six temperatures (minimum 600 °C and maximum 1160 °C). In addition, various characteristics of the BSS powder were also tested.

## 2. Materials and Methods

The Fe-Cr-Ni austenite with boron in accordance with Standard 304B7 of ASTM A887(2004, ASTM International, West Conshohocken, PA, USA) powder was obtained by atomization equipment, (Advanced Technology & Materials Co., Ltd., Zhuozhou city, China) as shown in Figure 1: Fe-B, Fe-Cr and other raw materials were put into the melting furnace (Advanced Technology & Materials Co., Ltd., Zhuozhou city, China) to melt and then were ejected, and the furnace was filled with protective argon gas so that the oxygen content of powder was low, approximately 200 ppm. The gas atomization powder obtained was passed through the 74 μm sieve, as shown in Figure 1. The powders were mostly spherical in shape, as shown in Figure 2. The elemental contents of boron, chromium, nickel etc. are shown in Table 1. The powder particle size distribution was tested by light scattering according to ASTM B822, as shown in Figure 3, which was the percentage of the volume content of the different particle sizes of the powder. Bulk density of powder was 4.53 g/cm$^3$, tested according to ISO 3923-1; tap density of powder was 5.15 g/cm$^3$, tested according to ISO 3953.

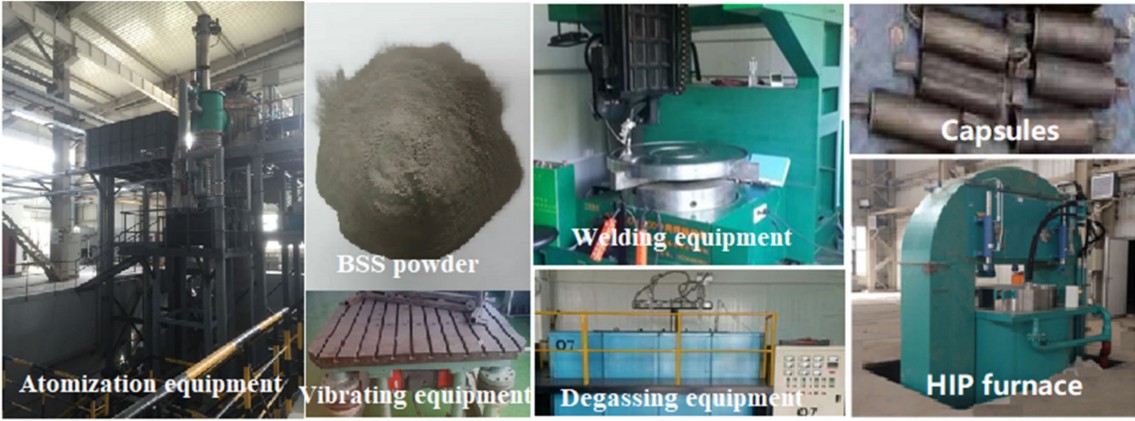

**Figure 1.** Production equipment, BSS (borated stainless steel) powder, and capsules photographs.

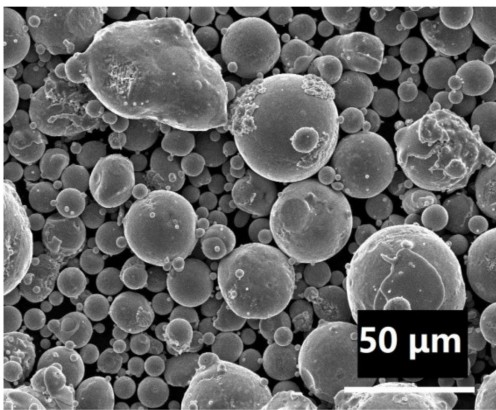

**Figure 2.** Metallographic of borated stainless steel (BSS) powder.

**Table 1.** Powder chemical composition in mass%.

| Element | B | C | Cr | Ni | Mn | Si | P | S |
|---------|------|-------|-------|-------|------|------|--------|--------|
| mass% | 1.86 | 0.020 | 19.30 | 14.10 | 2.00 | 0.66 | 0.0070 | 0.0051 |

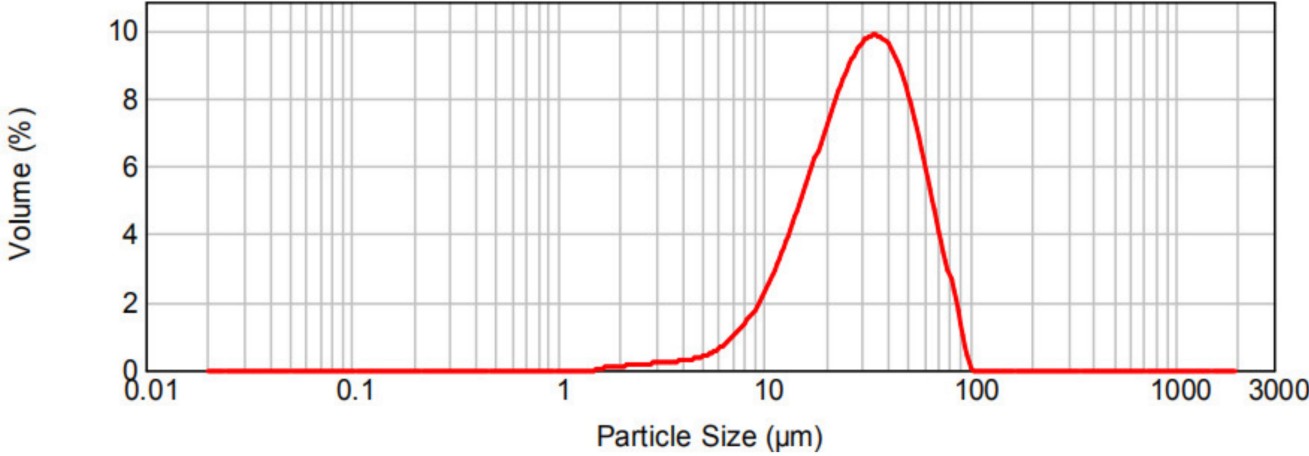

**Figure 3.** Powder particle size distribution.

The powder was placed in the capsules, which were welded by using Q235 steel according to Chinese Standard GB/T 700 (2006, Standardization Administration of the P.R.C., Beijing, China). The capsules were placed on the vibrating equipment to obtain higher tap density. Filled with BSS powder the capsules were heated to 500 °C in the atmospheric oven and degassed to $10^{-3}$ Pa to reduce oxidation of the powder. Held for one hour, the capsules were sealed. One capsule was placed individually in the HIP furnace and then heated up and one package was placed individually in the HIP furnace and then heated up and elevated pressure and held at sintering temperature and maximum pressure (110 MPa) for 2 h, as shown in Figure 4 for HIP process. The billets after sintering were cooled naturally in the HIP along with the furnace for approximately 6–10h and then taken out of the furnace below 150 °C. In this study there were six sintering temperatures: 600, 700, 900, 1000, 1100, and 1160 °C, designed to observe such characteristics such microstructure at different HIP sintering temperatures. Subsequent tests were carried out after the capsules were removed. In Figure 1 are shown photos of vibrating, welding, degassing equipment and HIP furnace.

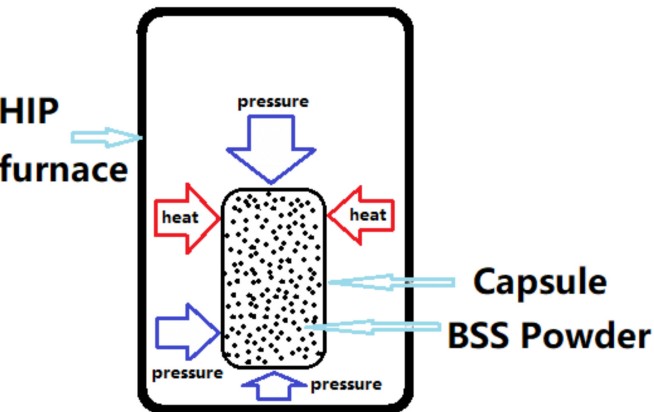

**Figure 4.** Drawing of HIP (hot isostatic pressing) sintering process.

Archimedes' law was used to test the density of the coupon at 1100 °C HIP; noted as ρT as the theoretical density, hot rolling reached 7.73 g/cm$^3$. The same method was used to test the density of HIP billets at different temperatures, with special attention to the need of applying petroleum jelly to the billets at 600 °C and 700 °C due to their connecting pores. Divided by the theoretical density ($\rho_T$ = 7.73 g/cm$^3$); noted as ρRi, the density was tested as relative density (where i is HIP sintering temperature).

Calculated by Thermo-Calc Software (2021a, Thermo-Calc Software, Solna, Sweden), the phases were compared with the actual phases. Coupons of different temperature HIP were evaluated by using quantitative X-ray diffraction (XRD) phase analysis. Microstructures were observed by scanning electron microscope (SEM) and transmission electron microscope (TEM) with energy-dispersive spectroscopic (EDS) analysis. Used for microstructure analysis, samples were polished and etched for microstructure analysis except for the sample at HIP 600 °C. Specimen at 600 °C was directly tested without polishing, and microstructure was analyzed by SEM, because it was not strong enough to be broken into pieces or powders during the sample preparation. Distributions and contents of B, Fe, Cr, and other elements at different locations were analyzed by TEM for temperatures: 900, 1000, 1100 and 1160 °C and SEM for at 700 °C, in which TEM samples could not be prepared for lack of sufficient strength. The grain size distribution and average grain size of the boron-containing phases at 900, 1000, 1100, and 1160 °C were counted and calculated. Strength and ductility of specimens from 900 to 1160 °C were tested according to standard ISO6892-1 (2019, International Organization for Standardization, Geneva, Switzerland), while specimens at 600 °C and 700 °C were excluded, because they didn't have sufficient strength and ductility.

### 3. Results

*3.1. Thermodynamic Calculation*

The thermodynamic calculation was carried out by Thermo-Calc Software (2021a, Thermo-Calc Software, Solna, Sweden) with TCFE10 database for the BSS powder under 110 MPa at the temperature from 500 to 1300 °C, as shown in Figure 5. Austenite, M$_2$B (boride), liquid phase, M$_{23}$C$_6$ (carbide) and ferrite were shown in the calculated equilibrium phase diagrams. The austenitic phase was stable in content when temperature exceeds approximately 550 °C, the M$_2$B phase was remained in content, the M$_{23}$C$_6$ phase was minimal and disappeared at approximately 800 °C, and the liquid phase appeared at approximately 1250 °C.

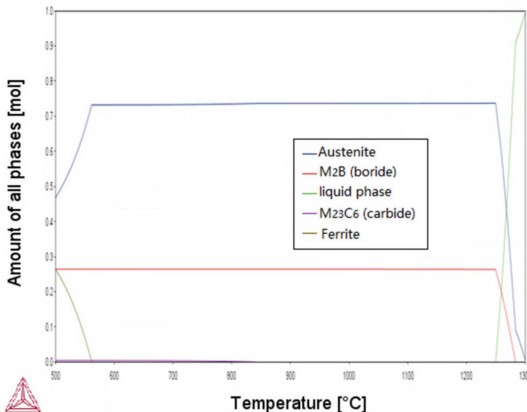

**Figure 5.** Calculated equilibrium phase content of BSS at 110 MPa and temperature from 500 to 1300 °C.

### 3.2. Quantitative X-ray Diffraction Phase Analysis

Figure 6 shows the X-ray diffraction (XRD) patterns of BSS powder and the alloy after HIP sintering at different temperature: 600, 700, 900, and 1000 °C. The BSS powder was supersaturated solid solution and the phase was austenitic. After undergoing HIP at 600 °C and 700 °C, alloy remained as single phase (austenitic) and no precipitation occurred; in comparison with Thermo-Calc software (2021a, Thermo-Calc Software, Solna, Sweden) calculations, no $M_{23}C_6$ and $M_2B$ phases appeared, the mainly reason was insufficient time of precipitation at lower temperatures. After experiencing HIP sintering at 900 °C, the boron phase was generated and the alloy consisted of two main phases: austenite phase and borides ($Fe_{1.1}Cr_{0.9}B_{0.9}$), and the latter was orthorhombic phase and generated above 900 °C of HIP temperature. $Fe_{1.1}Cr_{0.9}B_{0.9}$ and austenite phase were always present during the heat treatment and the proportion of them didn't change [5], while the $Fe_{1.1}Cr_{0.9}B_{0.9}$ phase proportion during HIP started to increase from zero at 700 °C. The XRD peaks of γ-Fe was broaden above 700 °C.

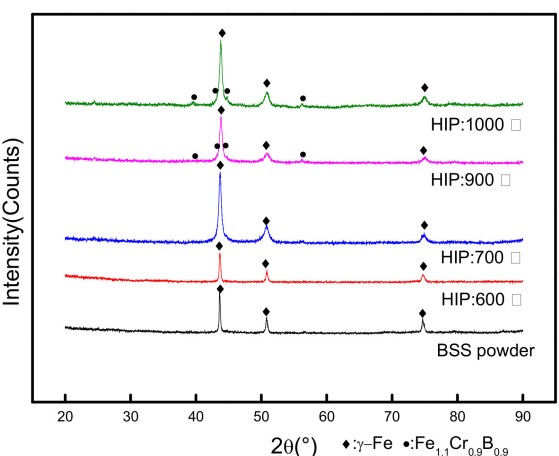

**Figure 6.** XRD (X-ray diffraction) of BSS (borated stainless steel) powder and alloy after HIP (hot isostatic pressing) sintering at different temperatures.

Figure 7 shows microstructure of BSS by HIP sintering at 1100 °C by the TEM observation with electron diffraction. It was demonstrated again that the boron phase was $Fe_{1.1}Cr_{0.9}B_{0.9}$, which was the dark phase in the figure.

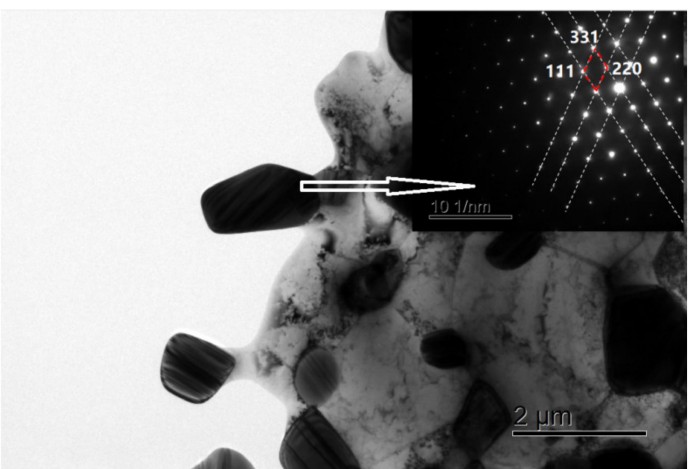

**Figure 7.** TEM (transmission electron microscope) image of BSS by HIP (hot isostatic pressing) at 1100 °C.

### 3.3. Relative Density

Figure 8 shows the variation curve of the relative density of alloy with increasing HIP sintering temperature. At 600 °C, the billet had density of 5.17 g/cm$^3$ (Relative density: 67%), which was essentially the same as the tap density, but the billet had already some strength. The trend was simulated in terms of high temperature yield strength of the capsule material by using JMatPro software (7.0, Sente Software, Guildford, UK), as is shown in Figure 9: the yield strength of Q235 steel at 600 °C might be higher than the pressure of HIP, so the powder in the capsule was similar to unpressured sintering, meanwhile there was essentially no increase in relative density compared to tap density. Without metallic luster, compact would crack if the capsule was removed, as is shown in Figure 10a.

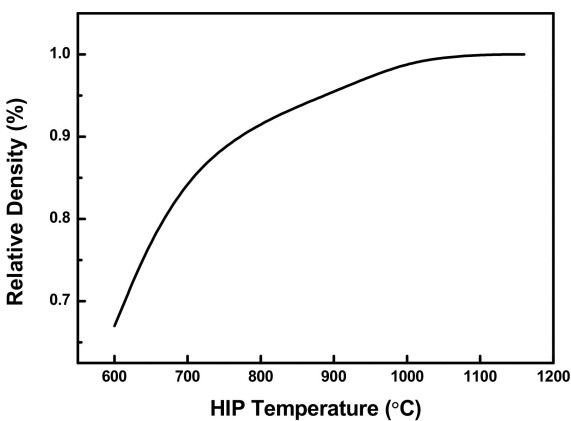

**Figure 8.** Relative density curve with HIP sintering temperature.

Increasing significantly at 700 °C, the relative density reached 88.3%. Meanwhile the billet took on a metallic luster and some strength, while the surface had many small visible pits, as is shown in Figure 10b. The relative density of the billet reached 95.4% at 900 °C and 99% at 1000 °C. The relative densities of the billets exceeded 99% after HIP sintering temperature stood at 1100 and 1160 °C. Billets at 900 °C above by HIP had the metallic luster but did not have visible pits, like in Figure 10c.

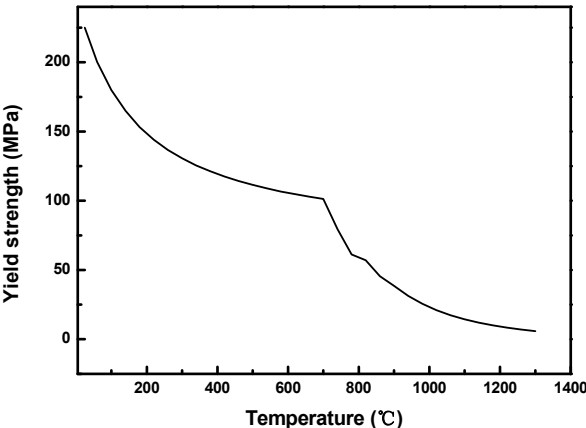

**Figure 9.** Temperature-dependent yield strengths of capsule material simulated by JMatPro Software.

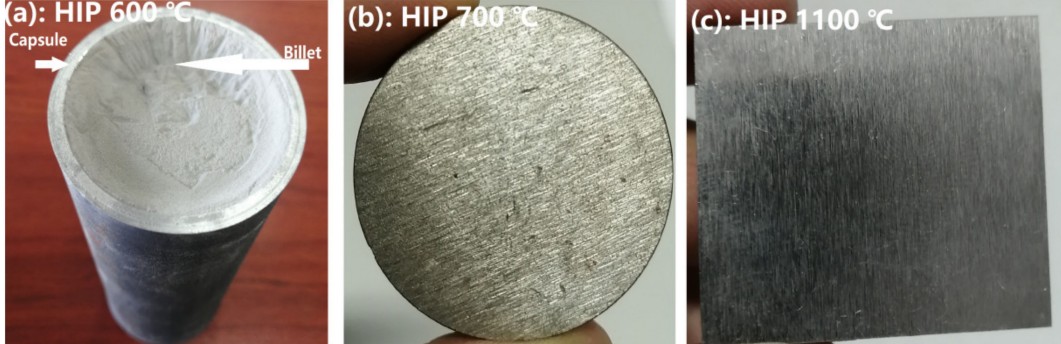

**Figure 10.** Photographs of samples by HIP at (**a**) 600 °C, (**b**) 700 °C, and (**c**) 1100 °C.

### 3.4. Metallographic Structure

The corresponding SEM characterizations are shown in Figure 11a,b for HIP sintering temperature of 600 and 700 °C, and the corresponding TEM characterizations are shown in Figure 11c–g for four other HIP sintering temperatures. Table 2 shows different elements content of the corresponding points of Figure 11c–g. After HIP temperature sintering stood at 600 °C, the BSS powder remained spherical rather than deformed, and no sintering neck appeared between the powder particles, as is shown in Figure 11a. When HIP sintering temperature stood at 700 °C, the larger grain size could reach 50 μm or even larger, and the smaller grain size was ranging between 10 μm and 30 μm, as is shown in Figure 11b. There were many micropores inside the grains, especially in the larger grains, and the micropores were distributed in a circular pattern at the position pointed out by arrow A. With greater magnification, the grains of HIP at 700 °C had different color microstructure inside the grains: one was dark in the form of dendrites, and the other was lighter in color and distributed between the darker areas, as is shown in Figure 11c.

When HIP sintering temperature stood at 900 °C, in austenite matrix were distributed a large number of smaller size grain boron-containing phases of tens to hundreds of nanometers, some smaller grains have coalesced (the area pointed by arrow B in Figure 11d), and two larger size grains (around 500 nm) had coalesced in a dumbbell shape (the area pointed by arrow C). Figure 11e shows there were boron-containing phase grains of different sizes: the smaller ones were below 100 nm, and the larger ones were several microns in length after HIP sintering temperature stood at 1000 °C. The austenite matrix gains contained smaller grains of boron-containing phase of approximately one hundred nanometers inside, and also larger grains of 500 nm. Located on the austenite grain edge, the boron-containing phase was large in size, reaching more than 500 nm; and the large-size boron-containing

phase particles were coalesced and grown. Figure 11f shows that when HIP sintering temperature stood at 1100 °C, there was no boron-containing phase inside of the austenite matrix grains. The large boron-containing phase grains were a few microns in size. There were diffusion zones for the elements in the small-sized boron-containing phase grains at the grain boundaries (the area pointed by arrow D). After heat treatment of the coupons processed at 1100 °C by HIP, the boron-containing phase grain size have changed. It became smaller when the heat treatment was at 900 °C and larger at 1200 °C. When the heat treatment ranged at 1000–1150 °C there was no big difference [5]. It was indicating that the boron-containing phase during the heat treatment process was unstable. Figure 11g shows that after HIP sintering temperature stood at 1160 °C, the grain size of austenite matrix phase and boron-containing phase grains remained basically unchanged from 1100 °C, but there were no small-sized boron-containing phase grains smaller than 1 micron. The structures with large boron content were present at the grain boundaries (the area pointed by arrow E).

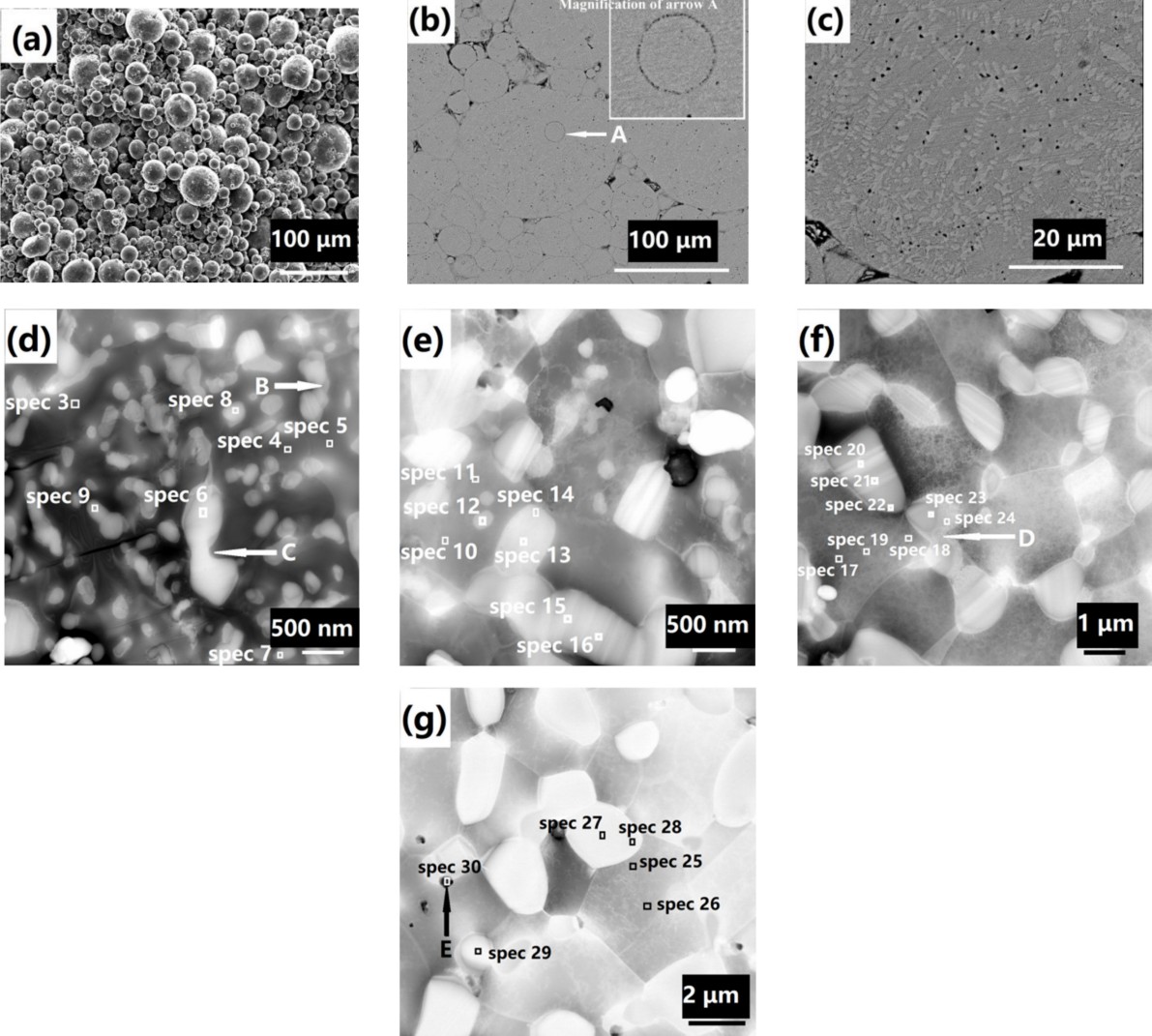

**Figure 11.** Microstructure and EDS (energy-dispersive spectroscopic) point analysis of BSS after HIP sintering at six temperatures: (**a**) 600 °C, (**b**) 700 °C, (**c**) 700 °C, (**d**) 900 °C, (**e**) 1000 °C, (**f**) 1100 °C, and (**g**) 1160 °C.

**Table 2.** Boron and other elements content of SEM (scanning electron microscope) and TEM (transmission electron microscope) EDS (energy-dispersive spectroscopic) points of BSS (borated stainless steel) (wt%).

| | Spectrum | B | Fe | Cr | Ni | Mn | Si |
|---|---|---|---|---|---|---|---|
| Figure 11c | 1 | - | 62.3 | 21.5 | 13.1 | 2.5 | 0.7 |
| | 2 | - | 64.8 | 16.9 | 15.1 | 2.0 | 1.2 |
| Figure 11d | 3 | - | 63.97 | 9.67 | 16.05 | 1.88 | 8.44 |
| | 4 | - | 66.96 | 9.97 | 16.89 | 2.11 | 7.79 |
| | 5 | - | 66.86 | 10.26 | 16.59 | 1.92 | 4.35 |
| | 6 | 3.10 | 33.86 | 59.94 | 0.47 | 1.98 | 0.90 |
| | 7 | 5.65 | 31.70 | 57.97 | 0.46 | 1.73 | 2.50 |
| | 8 | 1.98 | 32.82 | 60.53 | 1.11 | 1.97 | 1.45 |
| | 9 | 7.23 | 38.75 | 50.09 | 0.69 | 1.89 | 1.35 |
| Figure 11e | 10 | - | 68.63 | 11.23 | 16.30 | 2.00 | 1.84 |
| | 11 | - | 68.38 | 11.16 | 16.42 | 2.12 | 1.92 |
| | 12 | 4.08 | 53.65 | 30.08 | 9.64 | 1.94 | 0.61 |
| | 13 | 2.21 | 42.82 | 48.58 | 3.62 | 1.99 | 0.78 |
| | 14 | 4.16 | 47.12 | 39.94 | 6.14 | 1.92 | 0.70 |
| | 15 | 2.23 | 37.00 | 57.07 | 0.65 | 1.89 | 1.15 |
| | 16 | 1.85 | 37.68 | 57.32 | 0.67 | 2.02 | 1.85 |
| Figure 11f | 17 | - | 67.47 | 11.39 | 16.59 | 2.13 | 2.41 |
| | 18 | - | 68.09 | 11.87 | 16.68 | 2.08 | 1.29 |
| | 19 | - | 68.43 | 11.45 | 16.65 | 2.06 | 1.41 |
| | 20 | 2.58 | 37.79 | 56.55 | 0.76 | 2.04 | 0.28 |
| | 21 | 2.53 | 38.01 | 56.03 | 0.77 | 1.95 | 0.32 |
| | 22 | 3.65 | 38.29 | 54.79 | 0.81 | 1.87 | 0.59 |
| | 23 | 3.51 | 38.57 | 54.62 | 0.83 | 2.13 | 0.35 |
| | 24 | - | 58.67 | 27.43 | 11.12 | 1.97 | 0.80 |
| Figure 11g | 25 | - | 68.16 | 12.57 | 16.18 | 2.12 | 0.97 |
| | 26 | - | 68.41 | 12.48 | 15.90 | 2.16 | 1.06 |
| | 27 | 0.70 | 38.14 | 57.86 | 0.97 | 2.10 | 0.22 |
| | 28 | 3.06 | 40.77 | 52.87 | 1.27 | 1.86 | 0.17 |
| | 29 | 0.70 | 37.65 | 57.99 | 0.92 | 2.21 | 0.53 |
| | 30 | 36.31 | 23.51 | 35.68 | 0.53 | 2.63 | 1.34 |

When HIP sintering temperature stood at 900 °C, in austenite matrix were distributed a large number of smaller size grain boron-containing phases of tens to hundreds of nanometers, some smaller grains have coalesced (the area pointed by arrow B in Figure 11d), and two larger size grains (around 500 nm) had coalesced in a dumbbell shape (the area pointed by arrow C). Figure 11e shows there were boron-containing phase grains of different sizes: the smaller ones were below 100 nm, and the larger ones were several microns in length after HIP sintering temperature stood at 1000 °C. The austenite matrix gains contained smaller grains of boron-containing phase of approximately one hundred nanometers inside, and also larger grains of 500 nm. Located on the austenite grain edge, the boron-containing phase was large in size, reaching more than 500 nm; and the large-size boron-containing phase particles were coalesced and grown. Figure 11f shows that when HIP sintering temperature stood at 1100 °C, there was no boron-containing phase inside of the austenite matrix grains. The large boron-containing phase grains were a few microns in size. There were diffusion zones for the elements in the small-sized boron-containing phase grains at the grain boundaries (the area pointed by arrow D). After heat treatment of the coupons processed at 1100 °C by HIP, the boron-containing phase grain size have changed. It became smaller when the heat treatment was at 900 °C and larger at 1200 °C. When the heat treatment ranged at 1000–1150 °C there was no big difference [5]. It was indicating that the boron-containing phase during the heat treatment process was unstable. Figure 11g shows that after HIP sintering temperature stood at 1160 °C, the grain size of austenite matrix

phase and boron-containing phase grains remained basically unchanged from 1100 °C, but there were no small-sized boron-containing phase grains smaller than 1 micron. The structures with large boron content were present at the grain boundaries (the area pointed by arrow E).

Figure 12 shows the variation in grain average size with temperature and grain size distribution of boron-containing phase at 900, 1000, 1100, and 1160. At 900 °C the average grain size of the boron-containing phase was only 0.2 μm, with the smallest grain size detected being 60 nm; at 1060 °C the average grain size was 2 μm, the 100-fold increase in grain size. As can be seen from the grain size distribution diagram, there were two peaks at 1000 °C: the first was at a grain size of 0.15–0.2 μm, which was slightly larger than the peak of 900 °C at 0.1–0.15 μm; the second is at a grain size of 0.35–0.4 μm. When compared with Figure 11e, at 1000 °C, the boron-containing phase grains were mainly distributed in two places: inside the austenite grains and at the austenite grain boundaries; the grain size inside the matrix was relatively smaller than that at the grain boundaries, but grew larger than at 900 °C. The peak at 1100 °C was at 0.5–1.0 μm, while the peak at 1160 °C was at 2.0–2.5 μm. When compared with Figures 11f and 12b, there were no longer any boron-containing phase grains within the austenitic grains at 1100 °C, the grains at the grain boundaries were either large or small, the grains at 1160 °C were basically larger than 1.0 μm. Counted and averaged, the austenite grain sizes at 600 °C and 1160 °C were 17 μm and 4.6 μm, respectively.

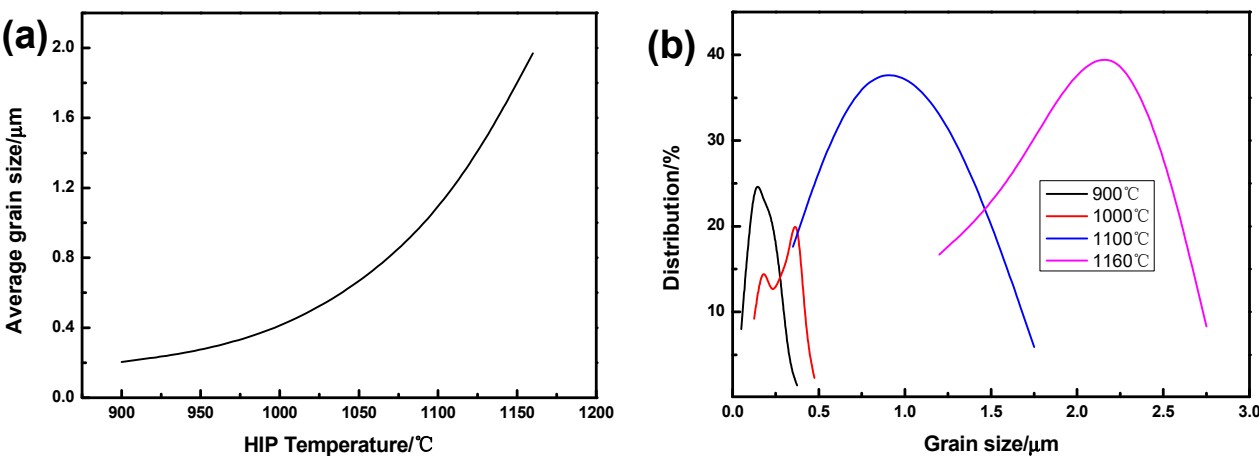

**Figure 12.** (**a**) Average grain size change at different HIP sintering temperature from 900 to 1160 °C; (**b**) grain size distribution at 900, 1000, 1100, and 1160 °C.

Table 2 summarizes the different element contents of SEM-EDS and TEM-EDS point scan analysis in Figure 11. When the HIP sintering temperature was 700 °C, the black area had a higher Cr of 21.5% and a slightly lower Ni of 13.1% (spectrum 1); the lighter area was the opposite, with a lower Cr of 16.9% and a slightly higher Ni of 15.1% (spectrum 2). The Cr content of austenite grains at 900 °C was approximately 10%, slightly less than they were at 1000 °C and 1100 °C (between 11% and 12%), and the value for both temperatures was slightly less than they were at 1160 °C (>12%). The contents of Fe, Mn and Ni in austenite remained essentially the same at different temperatures. The element Si at 900 °C was significantly higher than it was at other temperatures. The boron content of the edge of the grain was greater than the interior (for example, spectrums 13, 14 of Figure 11e, spectrums 21, 22 of Figure 11f and spectrums 27, 28 of Figure 11g). The Cr, Ni and Fe contents in the boron-containing phase of large-sized grains were similar at different temperatures: Cr at approximately 57%, Fe at approximately 37%, and Ni at a lower content of approximately 1%. The elemental diffusion transition region appeared at 1100 °C with 58.87% Fe, 27.34% Cr and 11.12% Ni (spectrum 24). The boron-rich zones occurred at 1160 °C, with the content of more than 36% (spectrum 30).

### 3.5. Mechanical Properties of BSS at Different HIP Sintering Temperature

Table 3 shows the mechanical properties of BSS after HIP sintering at four temperatures except for 600 and 700 °C. From 900 to 1160 °C, the tensile strength was similar, and the yield strength was the highest at 900 °C, followed by a decrease at 1000 °C and an increase to 1160 °C, reaching 483 MPa. The elongation was only 3% at 900 °C, increasing to more than 19% when temperature was at 1000 °C, then decreasing to 13% at 1100 °C and reaching 4% at 1160 °C with the temperature increases; the reduction of area follows the same rule. Without heating treatment, the tensile and elongation of coupon at 1000 °C by HIP without heat treatment were very close to the properties of heat treatment of BSS in Reference [5]. It was indicated that the same mechanical properties can be obtained by HIP at 1000 °C without heat treatment as that by HIP at 1100 °C HIP with heat treatment.

**Table 3.** Mechanical properties of BSS at different HIP sintering temperatures.

| HIP Temperature (°C) | Tensile Strength (MPa) | Yield Strength (MPa) | Elongation (%) | Reduction of Area (%) |
|---|---|---|---|---|
| 900 | 717 | 556 | 3 | 2 |
| 1000 | 776 | 341 | 19 | 18 |
| 1100 | 740 | 372 | 13 | 11 |
| 1160 | 783 | 483 | 4 | 6 |

Figure 13a–f shows the fracture morphology of HIP sintering from 900 to 1160 °C. The fracture morphology at 900 °C shows that there were many circular pits with white bright annular bands on the sides, as is shown in Figure 13a. Pointed by arrow F, the area was magnified with many small transgranular fractures, as shown in Figure 13b. In comparison with that of Figure 11d, these small (less than 1 μm) transgranular fractures should be the boron-containing phase, whose size was tens to hundreds of nanometers in Figure 11d. Analysis of Figure 13a,b showed that the boron-containing phase was mainly distributed on the edges of the round microstructure in Figure 13a (the bright circular bands of the edge); the boron-containing phase was also present inside the round microstructure, because the internal surface of the circular microstructure was not smooth in Figure 13b. Adjacent to the round microstructure, the small size of the boron-containing near the round microstructure was responsible for the lower elongation at 900 °C (3%); the boron-containing phase inside the round microstructure was responsible for the highest yield strength at 900 to 1000 °C (556 MPa). The fracture surface at 1000 °C was similar to that of 1100 °C, and the size of fracture grains ranged between below 1 μm to 5 μm. The absence of the round microstructure like 900 °C (size in the range of 10–50 μm) in the 1000 °C and 1100 °C fracture morphology indicated that the boron-containing phase had firstly grown and secondly was more evenly distributed in the austenite. A spherical shape of approximately 14 μm was found in the fracture surface of 1160 °C, which was surrounded by small transgranular fracture with surface smooth, as is shown in Figure 13e. Micro microstructure similar to aforesaid can be observed too in the fracture surface, which accounted for the elongation at 1160 °C, lower than that at 1100 and 1000 °C, as is shown in Figure 13f.

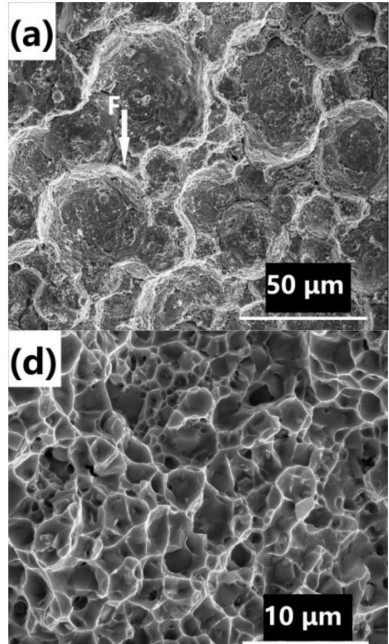
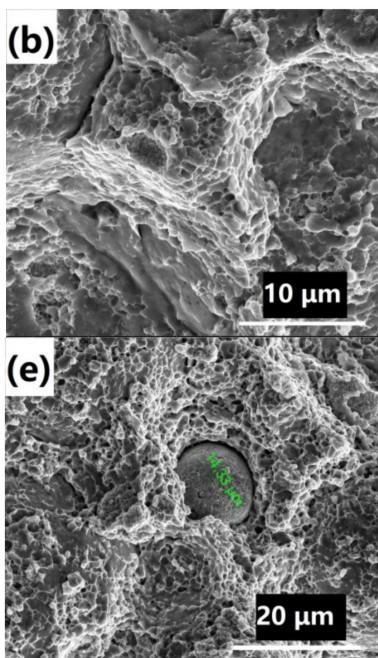
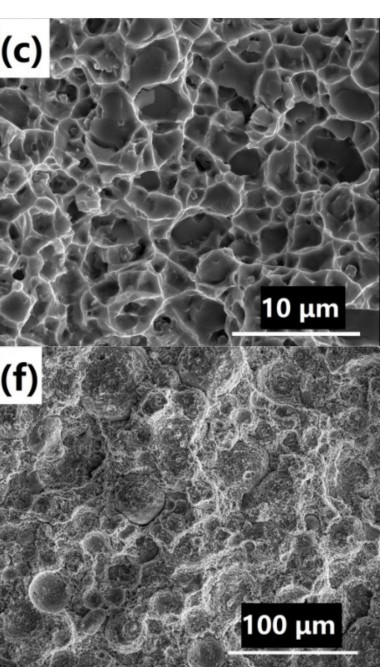

**Figure 13.** Fracture morphology of BSSs after HIP at (**a**) 900 °C, (**b**) 900 °C, (**c**) 1000 °C, (**d**) 1100 °C, (**e**) 1160 °C, and (**f**) 1160 °C of tensile tests.

## 4. Discussion

BSS microstructure and performance of the main influencing factors at different HIP sintering temperature: the low solubility of boron in steel, which causes the precipitation of supersaturated solid solution and the precipitation process due to the diffusion of boron and the different bonding of elements such as Cr, Fe, and Ni, different temperatures resulting in different grain sizes, boron-containing phase distribution, element distribution, etc.

The solubility of boron in $\alpha$-Fe and $\gamma$-Fe is very low, standing at 0.001–0.0025% below 900 °C [10]. The atomized BSS powder is the boron supersaturated solid solution, due to very fast cooling rate. In the densification process, the diffusion rate of boron increases with increase of temperature. The diffusion coefficient ($D$) of boron in $\gamma$-Fe is given by at 1223–1573 K [10]:

$$D = 2 \times 10^{-3} e^{-87,900/RT} \tag{1}$$

where $R$ is the universal gas constant (8.134 J K$^{-1}$ mol$^{-1}$) and $T$ is temperature (K). According to the above equation, the diffusion coefficient of boron at 1160 °C is 3.5 times that of 950 °C, then more times that of 900, 700, and 600 °C. Therefore, when HIP sintering time is the same, the higher the temperature the farther the diffusion will be. Boron segregation to austenite grain boundaries is considered to occur by a non-equilibrium segregation mechanism [29] and is mainly concentrated on the grain boundary, filled with defects, and boron phases generate with Fe, Cr, or Ni. The binding energy of boron and chromium is greater than that of the other elements in austenite [34], which is reason that the boron-containing phase is also chromium-rich region, in contrast to the Fe content, which is lower than the austenitic phase, and the nickel content, which is even lower than 1% in BSS.

Based on the experimental results, it is inferred that the nucleation and growth of the boron-containing phase are divided into four stages, as shown in Figure 14: I. precipitation of boron from supersaturated solid solution; II. generation of borides at grain boundaries or defects; III. generation of boron-containing phase; IV. growth of boron-containing phase.

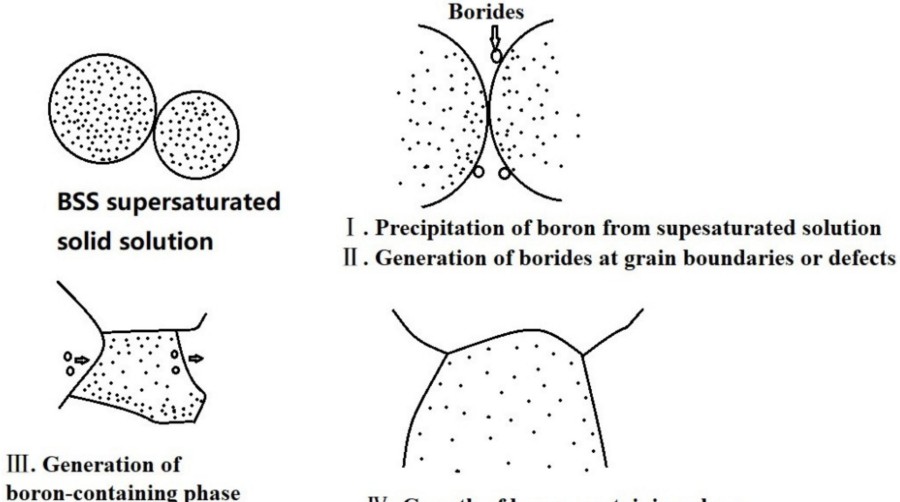

**Figure 14.** Drawing of boron-containing phase growth process.

The BSS powder is the boron supersaturated solid solution; as the HIP sintering temperature increases, boron is precipitated, causing a redistribution of Cr, Fe, and Ni elements, which is the reason for the difference between elements 1 and 2 of the spectrum in Figure 11c. At such lower temperature as 700 °C, this diffusion is short-distance, so that the darker and lighter colors in Figure 11c are distributed at intervals from each other. In addition, the broadening of the XRD peaks at 700 °C also indicate that the internal composition of the austenite solid solution becomes inhomogeneous: the broadening of the XRD peaks results from three factors: smaller crystallite size in nanocrystalline materials, inhomogeneous composition in the solid solution or alloy or more stacking faults, microstrain, and other defects in the crystal structure. The broadening is caused by the change in grain size as the temperature increases are negated (because the actual grain size is larger than powder size at 700 °C), the microstructure (Figure 11b,c) proves that the internal composition of the austenite became inhomogeneous, and the possible cause is the diffusion of the boron principle followed by vacancy defects causing the broadening. The presence of boron also accelerates the grain growth: pointed by the arrow A in Figure 11b, the ring-shaped micro pores are the traces left by the edges of the particles during the coalescence of the grains. Rao et al. investigated that the grain size did not grow significantly in comparison with the powder particle size after 1100 °C HIP using 304 stainless steel powders smaller than 180 μm [22]. However, in this study, abnormal grain growth was observed at 700 °C, indicating that boron can promote grain growth at lower temperatures.

In the first stage, as the HIP sintering temperature increases, boron atoms diffuse to the grain boundaries or defects such as micro pores due to their low solubility in the austenite matrix. In the second stage, boron forms borides at the grain boundaries with elements such as Fe and Cr. Such reaction also causes an imbalance of chemical potential and elements between austenite and boride.

In the third stage, since Cr and B bonding is greater than other elements, Cr in austenite diffuses to the boride position, while Ni near the boride diffuses into the austenite; Figure 11f shows the obvious element diffusion zone, and the proportion of Cr, Fe, and Ni in this region, which is different from that of austenite and boron-containing phase. With the increase of Cr content, more stable Cr-Fe solid solution is generated; the stability of Cr-Fe can be verified by the fact that the proportion of Cr to Fe (Table 2) in the boron-containing phase is essentially same at different temperatures. The Cr and Fe elements of the boride transform stable Cr-Fe solid solution with the Cr diffusing from the austenite, while the boron element diffuses again to the grain boundaries of the newly generated Cr-Fe solid solution due to the limited solubility, which is verified by the boron-containing phase in

which boron content of the grain boundary is greater than that of the grain interior, as is shown in Figure 11e,f and Table 2.

In the fourth stage, the boron element diffusing to the grain boundaries repeats the third stage process and then the grains grow. Apparently, the coalescence of small grains is another way of grain growth, as is shown in Figure 11c. Especially the processes of II, III and IV occur simultaneously and may even occur inside the austenite grains, as is shown in Figure 11e.

As is shown in Figure 11f, when the HIP sintering temperature increases to 1160 °C, more stable borides are generated at the grain boundaries, and this boride has higher boron content.

The effect of different microstructure caused by different HIP sintering temperatures determines the strength and elongation of the alloy: the high yield strength at the HIP sintering temperature 900 °C results from the precipitation strengthening effect of the particles formed by the boron phase precipitation, which improves the strength; while the low elongation mainly results from 95.4% relative density with many tiny pores producing the crack initiation. At 1000 °C, the relative density reaches 99%, and the boron-containing phase precipitated inside the austenite grains, giving alloy high tensile strength and elongation. At 1100 °C, the tensile strength and elongation are slightly reduced because the grain size of the boron-containing phase is larger than that is at 1000 °C. At 1160 °C, the high content of boron borides is generated around grain boundary and develops into crack initiation, causing a significant reduction in elongation.

## 5. Conclusions

In this study, the phases, microstructure, elements distribution, mechanical properties, and fracture morphology of BSS prepared by HIP were investigated at different temperatures from 600 to 1160 °C. The conclusion is drawn from the analysis of theoretical and experimental results that the key factor is the diffusion of boron and the bonding with other elements to determine the microstructure and properties of BSS during HIP densification, due to the low solubility of boron in the steel in the supersaturated solid solution. It was also inferred that the four-step model for the nucleation and growth of the boron-containing phase in BSS. The conclusions of presented work can be summarized as follows:

(1) Based on the experimental results and conclusion, the nucleation and growth mechanism of the boron-containing phase of BSS should be divided into four stages by HIP sintering at different temperatures: precipitation of boron from boron supersaturated solid solution, generation of borides at grain boundaries or defects, generation, and growth of boron-containing phase. The first step is controlled by the solubility of boron in the steel and the second to fourth steps are influenced by the diffusion distance of boron, the binding energy of boron with other elements and the solubility of boron in the Cr-Fe solid solution.

(2) The liquid stream ejected during the atomized powder making is broken up by argon, and the powder particles fall with the cooling rate of 1000 °C/s or more than, the obtained BSS powder is the supersaturated solid solution of boron. The relative density of BSS prepared by HIP at 600 °C is 67%. By SEM analysis, no coalescence and growth of the powder particles were observed. All these phenomena indicate that BSS powder prepared by HIP at 600 °C is in the "bonding stage" of PM sintering.

(3) HIP sintering at 700 °C, the short-range diffusion of boron elements, promotes the density of the billet and the abnormal growth of the grain, which is different from the ordinary stainless-steel materials, and the relative density of BSS can reach 88.7%. These phenomena show that HIP sintering at 700 °C is in the "sintering neck growth" stage of PM sintering. XRD analysis at 600 °C and 700 °C shows that the BSS has only austenitic phase, and the nucleation and growth of the boron-containing phase at this temperature is in the first stage with only the precipitation of boron.

(4) HIP sintering at 900 °C the boron-containing phase is generated and grows due to the increasing of diffusion coefficient of boron and other metallic elements and the alloy

contains two phases, boron-containing and austenite phases. According to EDS, the boron-containing phase is the Cr-rich, Fe-poor and the distribution of the boron is not homogeneous, the boron content of the smaller grains is higher than that of the larger grains and the boron content at the grain edges is higher than the internal content. The tensile strength is 717 MPa and the elongation is 3% due to the lower temperature, the fine grain size, and the relative density of 95.4%. The HIP sintering at 900 °C is controlled by diffusion and the nucleation and growth of the boron-containing phase at this temperature is in the second stage with precipitation of boron and growing of boron-containing phase at the grain boundaries or defects.

(5) HIP sintering at 1000 °C or higher, the solubility of elemental boron in austenite and the diffusion coefficient of the elements increase, and material transportation accelerates, so the density of alloy reaches more than 99%. The microstructure evolution is controlled by both solubility and elemental diffusion, the nucleation and growth of the boron-containing phase at this temperature is in the third and fourth stage. The highest elongation is 19% at 1000 °C and the tensile strength is 776 MPa, and the main reason is that the small grain boron-bearing phase distribute inside the austenite phase. The highest tensile strength can reach 783 MPa at 1160 °C and elongation is only 4%, the main reason is that the boron-bearing phase grain size is larger than the other temperatures.

Based on the phenomena and results of the above studies, the fine grain and fully dense alloy with higher strength and ductility for BSS may be obtained with HIP temperature between 900 and 1000 °C if the pressure is increased and the sintering time is extended.

**Author Contributions:** Conceptualization, T.W.; methodology, Y.P.; validation, X.Q. and Q.G.; formal analysis, Y.P.; investigation, Y.P. and T.W.; resources, T.W.; data curation, Y.P.; writing—original draft preparation, Y.P.; writing—review and editing, X.Q. and Q.G. All authors have read and agreed to the published version of the manuscript.

**Funding:** This research received no external funding.

**Institutional Review Board Statement:** Not applicable.

**Informed Consent Statement:** Not applicable.

**Data Availability Statement:** Data available in a publicly accessible repository.

**Conflicts of Interest:** The authors declare no conflict of interest.

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
