# Peer review of "Evolution of Microstructure and Elements Distribution of Powder Metallurgy Borated Stainless Steel during Hot Isostatic Pressing"

_metals, doi:10.3390/met12010019_

Round 1

Reviewer 1 Report

OK, my comments have been addressed now and I take it that this is sufficiently different from your previous publications.

It is a sound manuscript now.

Please have a quick look at the caption of Fig. 8 - apart from that this looks all good to me.

Author Response

Dear Review:
Thank you very much for quick, conscientious, professional and responsible comments on our paper. We have revised the manuscript according to your kind advice and detailed suggestions. Enclosed please find the responses to the referees. 

Reviewer 2 Report

Dear authors.

Thank you so much for your kind and elaborate reply to me. In my opinion your paper has improved clearly in this actual version. Congratulations for the work.

Best regards.

Author Response

Dear Review:
Thank you very much for quick, conscientious, professional and responsible comments on our paper. We have revised the manuscript according to your kind advice and detailed suggestions. We have modified some of the English style.

Best regards

Sincerely yours

Yanbin Pei

Reviewer 3 Report

The manuscript needs to be revised:

1.The annotation should reflect the relevance in the study of borated stainless steel (BSS), obtained by powder metallurgy, including spraying and sintering by hot isostatic pressing (HIP) at six different temperatures from 600 to 1160 ° C, with a boron content of 1.86 wt. % and it is imperative to clarify the novelty in the study of the mechanism of evolution of the microstructure of steel;

  1. add the brand of Thermo-Calc software to the keywords;
  2. the introduction should contain a clearly defined research goal and work objectives;
  3. In the main part of the manuscript, it is imperative to clarify the details of the experiment: provide a photograph of the installation and photographs of the samples obtained;
  4. Draw up figure 7, 8 and 10 in the same scientific style: eliminate an unnecessarily dense coordinate grid, repetition of the number zero on both axes and cumbersome inscriptions along the coordinate axes;

6.Rewrite the conclusions in the following structure: describe new details of the evolutionary mechanism of the microstructure of borated stainless steel (BSS) obtained by powder metallurgy, including spraying and sintering by hot isostatic pressing (HIP) at six different temperatures from 600 to 1160 ° C, with a boron content 1.86 wt% and add perspective for further research.

Author Response

Dear Review:
Thank you very much for conscientious, professional and responsible comments on our paper. We have revised the manuscript according to your kind advice and detailed suggestions. Enclosed please find the responses to the referees.

Best regards

Sincerely yours

Yanbin Pei

Round 2

Reviewer 3 Report

Ok

This manuscript is a resubmission of an earlier submission. The following is a list of the peer review reports and author responses from that submission.

Round 1

Reviewer 1 Report

The manuscript is characterized by many form errors. It seems it has not been read carefully by the Authors before submission.

In the following lines, some examples:

Figure 2: there is the word " borides" floating in the graph

Figure 3: X-axis is missing

Figures 4 and 5 and 6: marker and labels are not visible

Conclusions: conclusions can not start with a bullet point

Author Response

Dear reviewer:
Thank you very much for your attention and evaluation and comments on our paper “Evolution of Microstructure and Elements Distribution of Powder Metallurgy Borated Stainless Steel during Hot Isostatic Pressing”. We have revised the manuscript according to your professional, conscientious and careful advices and detailed suggestions. Enclosed please find the responses to the referees.

Best regards

Sincerely yours

Yanbin Pei

Reviewer 2 Report

The manuscript entitled: Evolution of Microstructure and Elements Distribution of PM Borated Stainless Steel during Hot Isostatic Pressing deals with the powder metallurgical processing of borated stainless steel. I have the following concerns with the manuscript.

  • Figure 2  Y-axes legend unit is missing. 
  • Figure 2 - Probably the borides are wrongly indexed. All the peaks should be indexed directly in Fig. 2.
  • Why does the XRD peaks get broaden when the HIP temperature increases?
  • Figure 3 - X-axed legend unit is missing.
  • The number is Table 2 has no units.
  • The scale bars in Figure 4 are hardly readable. Also the text in Figure 4.
  • It may be better if the same magnification images are used in Figure 4.
  • Scale bars in Figure 5 is hardly readable.
  • A strong scientific dicusssion correlating the the boron diffusion is missing.
  • Typos in the manuscript needs attention. For instance 776MPa should be written as 776 MPa.
  •  

Author Response

(The authors gave the same response as above.)

Reviewer 3 Report

Thank you for submitting this work to "Metals" - it is an excellent fit and it is a sound study.
Just like your previous study published in "Materials" - https://doi.org/10.3390/ma14164646 - I see some differences - to be able to accept your manuscript to metals - please go through the whole manuscript one more time and make sure you really presenting some new material, the first sentence is already suspitionsly similar...!

More feedback:
-please provide all authors emails - see MDPI template
-PM - for powder metallurgy? - I am not sure you need this in the title - and you should explain it more in your introduction

-you are not modelling BSS production but actually doing it - I think your literature review should rather focus on other experimental work and show what additional research is possible.
-Please also mention your own previous work here (https://doi.org/10.3390/ma14164646, etc.) and show how this paper is different

-results and experiments are good - really make sure it is different from your other study otherwise we cannot accept it
-conclusions can be extended - this is rather short

Author Response

(The authors gave the same response as above.)

Reviewer 4 Report

Best regards.

Author Response

(The authors gave the same response as above.)

Round 2

Reviewer 3 Report

-

Author Response

Dear Reviewer:
Thank you very much for your careful and professional review and comments. on our paper.  We had the manuscript revised by an English professional according to
your kind advice

Best regards

Sincerely yours

Yanbin Pei

Reviewer 4 Report

Dear authors.

Thank you so much for your kind and elaborate reply to me. In my opinion it has been a good idea to rewrite all those paragraphs, and several sentences, that was just a copy of your previous work in Materials. With relation to this last work, I have read it again and I am not able to find that the HIP temperature was only one (1100ºC) as you pointed to me. But this is another history, and it is not the topic here! Anyway, here we are. In my opinion the appearance of the present work has been remarkably improved.

I have found three minor remarks.

  • In table 1, in the header, you have change mass% by wt% in the last version. Inside the table the quantities are indicated in mass%. Which one is correct? (in your previous work of Materials the quantities are in mass%).
  • The term unfortunate term big ball is still inside figure 5.
  • Not all the references are in the Metals format.

Best regards.

Author Response

Dear Reviewer:
Thank you very much for your careful and professional review and comments. on our paper. We have revised the manuscript according to your kind advices suggestions.

Best regards

Sincerely yours

Yanbin Pei

Please find the following Response to the comments:

Response to the Reviewer’s comments
